# Inertial Sensor-Based Instrumented Cane for Real-Time Walking Cane Kinematics Estimation

**DOI:** 10.3390/s20174675

**Published:** 2020-08-19

**Authors:** Ibai Gorordo Fernandez, Siti Anom Ahmad, Chikamune Wada

**Affiliations:** 1Graduate School of Life Science and Systems Engineering, Kyushu Institute of Technology, 2–4 Hibikino, Wakamatsu-ku, Kitakyushu 808-0196, Japan; wada@brain.kyutech.ac.jp; 2Malaysian Research Institute of Ageing (MyAgeing™), Universiti Putra Malaysia, Selangor 43400, Malaysia; sanom@upm.edu.my

**Keywords:** instrumented cane, inertial measurement unit, gait, falls, deep learning, edge computing

## Abstract

Falls are among the main causes of injuries in elderly individuals. Balance and mobility impairment are major indicators of fall risk in this group. The objective of this research was to develop a fall risk feedback system that operates in real time using an inertial sensor-based instrumented cane. Based on inertial sensor data, the proposed system estimates the kinematics (contact phase and orientation) of the cane. First, the contact phase of the cane was estimated by a convolutional neural network. Next, various algorithms for the cane orientation estimation were compared and validated using an optical motion capture system. The proposed cane contact phase prediction model achieved higher accuracy than the previous models. In the cane orientation estimation, the Madgwick filter yielded the best results overall. Finally, the proposed system was able to estimate both the contact phase and orientation in real time in a single-board computer.

## 1. Introduction

Falls are among the most common causes of mobility impairment in the elderly [1]. Falls also reduce the quality of life, cause long-term injuries and pain, and sometimes lead to death. Worldwide, it is estimated that about 30% of people aged over 65 years will fall at least once per year [2]. This proportion depends on the country and world region; for example, in Japan, approximately 20% of the elderly population will fall at least once per year [3], whereas in Malaysia the percentage increases to 32.8% [4]. Although this trend might be caused by a difference in the methods for surveying falls, it is thought that the difference in the fall incidence might be caused by dietary and lifestyle habits, which are also related with a higher life expectancy [5]. The estimated fall percentages translate to millions of people worldwide. Furthermore, as societies are aging, the increasing trend in number of fallers is expected to continue over the coming years and decades. 

Following the above trends, the number of fall-related studies has also increased over the past few years. Most of these studies have focused on fall detection with the objective of reducing the adverse effects of falls. A system that detects fall occurrences can generate an alarm to caregivers or doctors, who can provide rapid treatment that supposedly reduces the negative consequences of falls [6]. Although immediate detection of a fall is important, systems that can estimate the risk of a fall are demanded for fall prevention. However, in a recent study of 300 papers on sensor development for fall detection and prediction in cane users [6], only five papers from two different groups had developed fall prevention systems. Moreover, both groups proposed robotic canes with multiple sensors and actuators. Fall prevention for normal cane users has not been reported.

Hence, our research focuses on the development of an instrumented cane that estimates the risk of fall in real time and provides feedback when there is a high fall risk. In a high-risk situation, the proposed system would alert the user to rest or contact a caregiver for assistance, rather than continuing to move. The sensors are placed on a single-tip cane because cane users are common among the elderly population; moreover, placement on the cane is less burdensome than placement on the body. The proposed system could estimate the fall risk by extracting the gait and balance parameters, which are well-known indicators of fall risk.

As an example, balance impairment is considered to greatly increase the fall risk [7]. Balance refers to the ability of regulating the position of the center of mass (CoM) within the base of support (BoS) to maintain equilibrium [8]. Balance is lost when a sudden movement or external perturbation displaces the CoM outside the BoS [9]. The fall risk is particularly increased during the single-leg phase of the human gait, when the BoS is reduced to the surface of the foot in contact with the ground. For this reason, many elderly individuals are provided with assistive devices such as walkers and walking canes, the latter being the most commonly prescribed mobility aid worldwide [9,10]. However, incorrect usage reduces the effectiveness of assistive devices. For instance, cane users might wrongly time the contact phase of the cane with the ground. Therefore, by measuring the correct timing of the cane–ground contact phase, we could acquire useful information on the effectiveness of canes to prevent falls.

As most falls occur during walking [11], fall risk is significantly associated with gait-related parameters such as stride variability [12]. To analyze the gait of cane users, researchers have employed parameters such the weight bearing on the cane [13] or the orientation of the cane [14]. Cane orientation has also been used as an estimator of fall risk in comparison studies of mobility-impaired patients and healthy elderly [15] and in fall detection of cane users [16]. Most of the studies used different types of algorithms to estimate the orientation from the data of an inertial sensor attached to the cane. However, although estimating the orientation gives useful information, the efficacies of different algorithms in estimating the cane orientation from inertial sensor data have not been compared.

In this study, we propose a real-time system that estimates the ground-contact phase and the three-dimensional (3D) orientation of a cane from inertial sensor data. Using the information of both the ground-contact phase and the cane orientation of the cane could be used to extract gait- and balance-related parameters for real-time estimation of fall risk. We first propose a deep learning model that improves the estimation of the ground-contact phase, in combination with a sliding voting window that improves the temporal accuracy of the system. Next, we compare the performances of various 3D cane orientation algorithms in the literature, using the data of an optical motion capture system as reference. Finally, although the proposed system only uses the data from one inertial sensor attached to the cane, during the experiments we employed multiple inertial sensors for analyzing the effect of sensor position on the accuracy of the proposed system.

## 2. System Overview

The proposed system consists of two components as shown in Figure 1. First, the contact phase detection block receives the inertial sensor data, and detects the instances of cane–ground contact. Next, the 3D orientation of the cane is calculated from the inertial sensor data and the previously estimated contact phase. In the following subsections, the system elements are explained in detail.

### 2.1. Ground-Contact Phase Estimation

By detecting whether the cane is contacting or not contacting the ground, we can analyze when the cane is increasing the BoS area and thus reducing the risk of fall. Moreover, by analyzing the duration of the ground contact, we can extract the cane stride time, cane cadence, and duration ratio of the contact and noncontact phases. These parameters provide useful information on the temporal variability of the gait. In addition, knowing when the cane is contacting the ground, it can help to estimate additional parameters such as the cane orientation and cane displacement. For example, Dang et al. [17] used the ground contact time to estimate the walking distance of a cane equipped with an inertial sensor. Their walking distance estimation method used the contact phase to improve the accuracy by using an inverted pendulum model when the cane was in contact with the ground.

The cane contact phase has been detected by various methods. One of the most common methods is to use a force sensor installed on the tip of the cane to detect when the reaction force exceeds a specific threshold [18]. However, force sensor-based contact phase estimation methods suffer from various limitations. First, when the force sensor is placed on the tip of the cane, the reduced friction between the cane tip and the ground could increase the slipping risk. Other researchers [14,19] modified the cane in order to place the force sensor between the tip and the shaft of the cane. Although this design removes the increased slip risk, the cane must be modified for placing the sensor inside the shaft, which increases the cost and complexity of the cane and could potentially affect the user’s gait.

In contrast, as inertial sensors are small and lightweight, they can be easily attached to any point of the cane. Previous inertial sensor-based cane contact phase estimation algorithms can be categorized into three groups: threshold-based methods [17], event-based methods [20,21,22], and machine learning methods. Our proposed method belongs in the third category.

The following subsections discuss the three categories of contact phase estimation, and present the proposed Deep Learning contact phase estimation model.

#### 2.1.1. Threshold-Based Methods

Threshold-based methods are the least complex and fastest methods for contact phase estimation. These methods use simple decision thresholds to estimate whether the cane is contacting the ground or moving in the air. For example, Dang et al. [17] compared the absolute value of the linear acceleration and the module of the angular velocity of the cane against their respective threshold values as follows:(1)$$\left|\|y_{acc}\|-g\;\right|\leq Thress_{acc},\;\|y_{gyro}\|\leq Thress_{gyro},$$
where $y_{acc}$ and $y_{gyro}$ represent the 3D accelerations and angular velocities obtained from the inertial sensor, respectively, and *g* denotes gravity.

As long as both the absolute value of the linear acceleration is at or below the acceleration threshold ($Thress_{acc}),$ and the module of the angular velocity is at or below the angular velocity threshold ($Thress_{gyro}$), Dang et al.’s algorithm estimates that the cane is contacting the ground. The threshold values were manually determined by analyzing the captured data. For a single-top cane, the acceleration and the angular velocity thresholds were fixed at $0.2\frac{m}{s^{2}}$ and $\;0.3\frac{rad}{s}$, respectively. In estimating the contact phase, the authors considered that both the acceleration and angular velocities of the cane are smaller during the ground-contact phase than during the air-moving phase (see Figure 2).

However, threshold-based methods are sensitive to noise. Inertial sensors are highly unstable, and sudden peaks above the threshold can trigger false estimations. Accelerometer signals are particularly susceptible because even small impacts can generate sudden destabilizing peaks. For this reason, although threshold-based methods can estimate the phase in real time with low-cost microcontrollers, their relatively low accuracy and high noise sensitivity limit their usability in a real-life estimation.

#### 2.1.2. Event-Based Methods

Event-based methods detect gait events from the characteristic shapes of sensor signals during ambulation. For example, Sprint et al. [20] detected the cane landing (initial contact, IC), cane mid-swing (MS) and cane take off (end contact, EC) from the angular velocity signals of the cane in the sagittal plane. In Figure 3, the positive peak in the sagittal angular velocity corresponds to a cane MS event, and the negative peaks correspond to IC and EC events. Once the IC and EC events are detected, the ground-contact phase can be estimated because it occurs between the IC and EC events, whereas the no-contact phase occurs between the EC and the next IC event.

Sprint et al.’s [20] method first searches for the positive peak in the angular velocity signal, which marks the cane MS event. It then seeks the negative peaks immediately before and after the positive peak, which identify the EC and IC events, respectively. The positive peak is detected first because the magnitude of the angular velocity peak is higher during the MS phase than during the magnitude of the IC and EC peaks, so the MS peak is the most easily detectable peak of the three.

Similarly, Gill et al. [21,22] detected the same gait events by combining the angular velocity with the output signal of a strain gauge attached to the cane. Their method detects an MS event as the positive peak in the angular velocity as in [20], but detects the IC and EC events by analyzing the zero-crossing points of the strain gauge output rather than the angular velocity.

Despite their high detection accuracy for continuous gaits, event-based detection methods might miss the contact phase if the signal shape changes due to gait impairment or for activities other than walking. Moreover, the reported event-based methods for cane contact phase estimation are not suitable for real-time applications, because the IC and EC peaks cannot be detected until the MS peak has been detected. Therefore, such systems cannot correctly detect the pre-MS contact phase in real time.

#### 2.1.3. Machine Learning Methods

With the development of machine learning methods, an increasing number of studies have applied machine learning models to applications using inertial sensor data, such as activity recognition [23,24], fall detection [25,26], and gait-phase detection [27,28]. Recently, Caro et al. [29] used a single hidden layer Neural Network (NN) on the data from force sensors on the tip of a cane to estimate the cane contact phase. Because the neural network adapts to multiple gait conditions, it improves the estimation accuracy of the cane contact phase compared to a finite state machine method.

Here, we propose a cane contact phase estimation model using a small size convolutional neural network (CNN), which delivers higher accuracy than the neural network with a single hidden layer. The CNN is a type of artificial neural network inspired by the visual cortex [30], in which each cortical area in the information transmission path is more specialized than the previous area. Unlike neural networks with a single hidden layer NN, CNN are able to learn how to extract more detailed features as its depth increases. Being based on the visual cortex, CNN models are commonly used in image-based applications, but have also been used in inertial sensor-sensor based applications such as activity recognition [31] and the detection of Parkinson’s disease-related events [32,33]. In such cases, instead of using an image input, the input is typically replaced by a 2D (*n* × *m*) matrix containing *n* samples from *m* inertial sensor signals.

The initial part of an inertial sensor-based CNN model is built using 1D convolutional layers followed by a max-pooling layer, which are repeated multiple times sequentially. These convolution layers help the model to extract features from the inertial sensor. Increasing the number of 1D convolution layers increases the detail in the extracted features and the complexity of the patterns that can be modeled. The final part of a CNN model normally consists of fully connected (FC) layers that take the features extracted from the convolutional layers, and learn the relationships among the features for predicting the desired output.

Our proposed model, due to the requirement to perform the estimation in real time even in devices with low computational resources, we adopted a modified CNN model with fewer and smaller-sized layers than most CNN models in the literature. Here, the layer size was reduced to increase the inference speed. The proposed model inputs a 2D matrix (*n* × 6), where the *n* samples are provided by a 3-axis accelerometer and a 3-axis gyroscope. At each instant, the last *n* samples are extracted from the inertial sensor data by a sliding window. The input data are passed to a 1D convolutional layer followed by a max-pooling layer. Next, the features extracted from the 1D convolutional layer are passed to an FC layer. Finally, the output of the FC layer is passed to a final FC layer with a Softmax activation function, which estimates the current contact phase. The proposed model outputs one of two labels: “ground contact” (when the cane is contacting the ground) or “no contact” (when the cane is moving in the air).

### 2.2. Cane Orientation Estimation

The next step of our proposed system calculates the 3D orientation of the cane using the data from the inertial sensor and the contact phase estimated by the previous step. Estimating the cane orientation from inertial sensor data is commonly reported in application studies such as gait assessment [14,15], fall detection [16,34], and lower-limb exoskeleton control [18]. However, an accuracy comparison of the different cane orientation estimation methods has not been conducted. In the following sections, we explain these various methods while the estimation accuracies of cane orientation by each method are compared in Section 4.2.

#### 2.2.1. Orientation from the Gravity and Magnetic North

The simplest method to calculate the cane orientation is to estimate it directly from the accelerometer and magnetometer signals [15,34]. These methods recognize that when the cane is not moving, the accelerometer measures only the gravitational acceleration, which always points downwards. Therefore, the roll and pitch angles of the cane are respectively estimated as follows:(2)$$roll=\tan^{-1}\left(\frac{a_{ML}}{\sqrt{a_{V}^{2}+\;a_{AP}^{2}}}\right),\;pitch=\tan^{-1}\left(\frac{a_{AP}}{\sqrt{a_{V}^{2}+\;a_{ML}^{2}}}\right),$$
where $a_{AP}$, $a_{ML}$ and $a_{V}$ represent the accelerations obtained by the inertial sensor in the anterior–posterior, mediolateral and vertical directions, respectively.

Although the yaw angle cannot be directly estimated from the accelerometer, it can be estimated by knowing that the Earth’s magnetic field always points north. Therefore, it is computed from the magnetometer data as follows:(3)$$yaw=\tan^{-1}\left(\frac{-m_{E}}{m_{N}}\right),$$
where $m_{E}$ and $m_{N}$ represent the transformed magnetometer data in the east and north directions, respectively. They are respectively calculated as follows:(4)$$m_{E}=m_{AP}\cos\left(pitch\right)+\;m_{V}\cos\left(pitch\right),\;m_{N}=m_{AP}\sin\left(roll\right)\sin\left(pitch\right)+\;m_{ML}\cos\left(roll\right)+m_{V}\sin\left(roll\right)\cos\left(pitch\right),\;$$
where $m_{AP}$, $m_{ML}$ and $m_{V}$ represent the magnetometer signals in the anterior–posterior, mediolateral and vertical directions, respectively.

As this orientation estimation requires few computations, this method is suitably fast for real-time applications. The accelerometer uses the gravitational acceleration for computing the roll and pitch angles. However, when the cane moves, the acceleration caused by the movement of the cane or impact of the ground is added to the gravitational acceleration. Thus, the accuracy of this method declines for activities where the cane is moving. Moreover, the noise of the inertial sensors affects the orientation estimation, and must be removed by smoothing. Similarly, the yaw angle estimated from magnetometer data can be affected by magnetic disturbances generated by metallic objects or electronic devices. Consequently, the accuracy of these methods is insufficient for the estimation of cane orientation in real life.

#### 2.2.2. Kalman Filter

On the other hand, Kalman filter-based [35] methods have been developed to improve the orientation estimation from inertial sensor data [36,37,38]. The inertial sensor data are fused with the model of the system to find the system state that minimizes the mean squared error by a recursive procedure [39]. Consequently, these methods can estimate the orientation, position, and other parameters with high accuracy, and are popular choices in multiple applications. Particularly, in the case of the cane orientation estimation, Culmer et al. [14] used a Kalman smoother to estimate the pitch and roll angles of a cane during walking from 6-axis inertial sensor data. Different from a Kalman filter, a Kalman smoother requires that all data are available. Therefore, it is suitable only for post-processing analysis and cannot be used in real-time applications.

Similarly, Dang et al. [17] applied an indirect Kalman filter to the data of a 9-axis inertial sensor attached to a cane to estimate both the 3D orientation and 3D displacement of the cane. The indirect and direct Kalman filters [38,40] differ by estimation of the orientation in the direct Kalman filter, whereas the indirect filter estimates the orientation error. Accordingly, the indirect Kalman filter has fewer state dimensions and a faster response than the direct Kalman filter [38]. In the indirect Kalman filter used by Dang et al. [17], the state variable consists of the errors in the estimated quaternion orientation, velocity, and position, as shown in Equation (5). Their algorithm updates the measurements during the ground-contact phase. For a single-tip cane, the measurement update is based on the equations of an inverted pendulum.
(5)xstd≜qereve∈ℝ9x1,
where qe, ve, and re represent the errors in the estimated quaternion orientation, velocity and position, respectively.

However, despite the high accuracy of Kalman filters, the main limitation of Kalman filter is the high computation cost due to their complexity. Consequently, they are inapplicable to real-time cane orientation using low power devices due to the required computation time. Ferdinando et al. [41] implemented a Kalman filter with simplified model equations in a microcontroller, and estimated the orientation from the inertial sensor data. As a result, the simplified model reduced the accuracy of the estimated orientation, and the system was sensitive to sensor noise.

#### 2.2.3. Madgwick Filter

To overcome the limitations of Kalman filters, Madgwick [42] presented a quaternion-based orientation estimation filter that achieves similar accuracy to a commercial Kalman filter but at low computational cost. The Madgwick filter behaves as a proportional controller with a filter gain *β* that controls the amount by which the accelerometer and magnetometer data correct the orientation estimated by the gyroscope. The magnitude of the (constant) filter gain is selected such that the filter minimizes the drift error while being robust enough to avoid the divergence caused by measurement errors in the accelerometer and magnetometer. Applying a gradient descent algorithm, the Madgwick filter fuses the gyroscope data with the orientation estimated by an accelerometer and magnetometer. Feng et al. [43] compared the execution times of the Madgwick filter and various implementations of Kalman filters, and reported a significantly faster calculation speed in the Madgwick filter.

Although the Madgwick filter is widely used in several applications, it has been rarely applied to estimate the orientation of a cane. Hassan et al. [18] applied a Madgwick filter to estimate the orientation of a single-tip cane to control a lower-limb exoskeleton. However, their application considers only the orientation of the cane in the sagittal plane. Moreover, the obtained cane orientation was only visually compared against the data of an optical motion capture system, but no quantitative performance was presented.

To the authors’ knowledge, among the many papers that have estimated cane orientation using inertial sensor-based algorithms, only one paper [14] has numerically validated the accuracy of the estimated orientation. In that study, the cane orientation was studied to investigate the effect of functional electrical stimulation, and only the roll and pitch accuracy were compared. Furthermore, the validation was performed on two measurements of one subject walking straight for 10 m. To address this deficiency, the present study numerically compares the accuracies of the different orientation estimation algorithms against the experimental results of multiple subjects performing multiple daily life activities.

## 3. Methodology

The proposed system was validated in two experiments. All participants of both experiments gave their informed consent before participating in the study. The study was conducted in accordance with the Declaration of Helsinki, and the protocol was approved by the Ethics Committee of Kyushu Institute of Technology (Approval ID: 19-07).

### 3.1. Instrumented Cane

In both experiments, participants used a single-tip cane equipped with five wireless inertial sensors (Logical Products, Japan), each containing a 3-axis accelerometer with a range of ± 5 g, a 3-axis gyroscope with a range of ± 1500 °/s, and a 3-axis magnetometer (not used in the contact phase detection) with a range of ± 1.3 Gauss. The five inertial sensors were placed throughout the cane at 15-cm intervals. The first sensor was placed below the handle of the cane, as shown in Figure 4. The proposed system uses the data of one inertial sensor, but five inertial sensors were used to analyze the effect of their position on the accuracy of the estimated parameters.

To validate the contact phase estimation, the reaction forces of the cane were measured by a force sensor (Tekscan, Boston, MA, USA) placed on the cane tip. The contact phase was labeled as “ground contact” when the reaction force exceeded 1 N, and “no contact” otherwise. The threshold value was decided based so that it had the smallest value above the noise in the force sensor when no force was applied. The force sensor signal was read by a wireless data logger (Logical Products, Fukuoka, Japan), which was also attached to the cane (see Figure 4). The data from the five inertial sensors and the data logger were obtained at 100 Hz and transmitted wirelessly to a computer for later processing. All inertial sensors were firmly attached to the cane with 3D printed parts to avoid the IMUs from shaking. Prior to each experiment, all IMUs were calibrated to reduce the effect of noise in the inertial sensor data. The force sensor was similarly calibrated before each experiment, and the relationship between the measured force in N and the measured voltage by the force sensor was then obtained.

### 3.2. Experiment 1: Cane Phase Estimation Evaluation

The objective of the first experiment was to validate the proposed cane contact-phase estimation model. Five healthy young participants (four males, one female, age 23.4 ± 1.52 years, height 171.5 ± 7.55 cm, weight 62.5 ± 9.95 kg, all statistics reported as mean ± standard deviation) performed different daily life activities while using the instrumented cane in a laboratory environment. Ten measurements from each participant were acquired while they performed seven daily life activities in a continuous sequence (see Table 1). Where possible, the order of the activities was randomly selected to avoid any influence of the activity order, which might affect the results. However, some of the activities were required to follow other activities; for example, the “sit to stand” movement must necessarily follow the “stand to sit” activity, and “walking upstairs” was always followed by “walking downstairs”. Before the experiment, each participant was taught the correct cane ambulation method for each activity, following the guidelines of the Health and Welfare Information Association (Japan). Moreover, the participants practiced the activities with the cane until they were proficient with its use.

The cane contact phase was estimated by a small size CNN (see Figure 5). The model was input with a (20 × 6) matrix corresponding to 20 samples from six signal (three each from the accelerometer and the gyroscope). The number of input samples (20) was selected because it provided a good trade-off between the accuracy and prediction speed of the model in preliminary experiments. Enlarging the window size could improve the model accuracy but increased the model size, thus increasing the computational time.

The experimental data were processed through a size-20 sliding window, selected to match the size of the input model. Therefore, for each sample *j*, the output yj contained the label of the cane contact phase at that instant, and the input xj  was given as
(6)xj= accXj−19accYj−19accZj−19gyrXj−19gyrYj−19gyrZj−19accXj−18accYj−18accZj−18gyrXj−18gyrYj−18gyrZj−18              …accXj−1  accYj−1   accZj−1   gyrXj−1  gyrYj−1  gyrZj−1   accXj        accYj         accZj         gyrXj        gyrYj         gyrZj         ∈ℝ20x6,
where accX, accY and accZ represent the signals of the 3-axis accelerometer and gyrX, gyrY, and gyrZ represent the signals of the 3-axis gyroscope.

A total of 338,316 samples were collected from each inertial sensor, and a total of 1,691,580 samples were obtained from all IMU sensors. The model was trained and tested using leave-one-out cross validation; at each fold, the data of four participants were used in training and the data of the remaining participant were reserved for testing.

The proposed CNN model contained four hidden layers (Figure 5). The first layer was a 1D convolutional layer with 8 filters of size 3 × 1 and a rectified linear unit (ReLU) [44] activation function. This layer processed the input of the model, and passed its output to a max-pooling layer of size 2. This was followed by a flatten layer that converted the output of the max pooling layer to a 1D array.

The model was completed by two FC layers. The first FC layer consisted of eight neurons with a ReLU activation function, whereas the final FC layer consisted of one neuron with a Softmax activation function, which predicted the class of the cane contact phase (“ground-contact” or “no contact”). The proposed model had 745 parameters, significantly fewer than the number of parameters in similar CNN models, enabling faster prediction speed. The model was implemented and trained using the Keras neural network library on a GTX 950M chip (Nvidia Corporation, Santa Clara, CA, USA), and the prediction time was computed in a Raspberry Pi 3B (Raspberry Pi Foundation, Cambridge, UK) single-board computer.

### 3.3. Experiment 2: Cane Orientation Evaluation

In the second experiment, we compared different cane orientation estimation methods. The five participants in Experiment 1 were joined by another five participants. The 10 participants (nine males, one female, age 23.8 ± 1.99 years, height 171.4 ± 5.42 cm, weight 63.6 ± 8.57 kg, all statistics reported as mean ± standard deviation) performed the activities designated in Experiment 1 using the same instrumented cane.

To validate the orientation estimated by the inertial sensor, the ground truth orientation was obtained by an optical motion capture system (Optitrack, Corvallis, OR, USA) equipped with eight infrared cameras. The ground truth orientation of the cane was estimated by tracking the positions of five markers placed on the cane, as shown in Figure 6. One marker was placed over IMU 1, another over IMU 4, and the remainder on the three perpendicular arms (one marker per arm) of a 3D-printed attachment. Each arm was 10 cm long.

Both the motion capture system and the sensors in the instrumented cane were sampled at 100 Hz. Moreover, to synchronize the sensors with the motion capture data, participants were requested to raise the cane rapidly before and after each measurement. Thereby, we could compare the acceleration peaks from the inertial sensor with those of the motion capture system. As in Experiment 1, all inertial sensors and the force sensor were calibrated prior to Experiment 2. All calculations and comparisons of the cane orientation estimation were performed in MATLAB 2020a (MathWorks, Natick, MA, USA) on the same computer as in Experiment 1.

As the measuring range of the optical motion capture system is limited, the activities were completed not in a continuous sequence, but were separately measured 10 times per activity. A total of 700 measurements were collected, yielding 413,419 samples for each inertial sensor.

## 4. Results and Discussion

This section presents the estimation results of the cane contact phase and cane orientation, and discusses their importance.

### 4.1. Cane Phase Estimation Results

First, the cane contact phase estimated by the proposed CNN model was compared with those of previous inertial sensor-based methods, namely, the threshold- [17] and event-based [20,21] methods explained in Section 2. Table 2 lists the results of the compared methods at each fold of the cross validation. The proposed model outperformed the existing models, both at each fold and on average. The accuracies of both previous methods were quite similar, with the threshold-based method performing slightly worse (82.27% accuracy) than the event-based method (84.33% accuracy). The average accuracy of the proposed model (95.46%) was 13.19% higher than in the threshold-based method, and 11.13% higher than in the event-based method.

In the threshold-based method, both threshold values ($Thress_{acc}$, $Thress_{gyro}$) needed to be modified from their original values [17] to 0.7. The accuracy of the threshold-based estimator with the original threshold values was only 52.31%. The low accuracy of this model with the original threshold values might be attributable to high noise in the inertial sensors of the present experiment, which can cause false ground-contact estimations. The decreased accuracy might also be explained by the nature of the current study activities; for instance, stair climbing was not included in the original study. If these additional activities have higher signal values during the “ground contact” phase, the smaller threshold values might not have been able to differentiate the two contact phase types.

To analyze the effect of the inertial sensor position on the accuracy of the proposed method, the test accuracies for each inertial sensor are compared in Table 3. IMU 4 yielded the highest accuracies in most of the participants and the highest average accuracy. However, after performing a Kruskal–Wallis nonparametric test with a significance level of 0.05, no significant differences were found among the results of the inertial sensors. This implies that the proposed model was independent of the inertial sensor placement on the cane.

Figure 7a compares the phases estimated by the proposed model with the ground truth contact phase. The estimated phase was unstable at certain times, in those cases, the estimated phase changed rapidly between “ground contact” and “no contact” even though there was no phase change in the ground truth phase. However, such rapid contact phase changes are not expected during normal use of the cane. To smooth the estimation of the phase, a voting window was applied to the output of the model.

The voting window contained the last 25 estimations of the proposed model, from which the current phase was decided by majority voting (Figure 5). If more than half of the samples in the voting window were classified as “ground contact”, the contact phase was set to “ground contact”; otherwise, the contact phase was set to “no contact”. Although the majority vote increased the accuracy only slightly (from 95.46% to 95.72%; see Table 2), it smoothed the model estimation, as shown in Figure 7b.

The instability in the phase estimation might have been solved using a different neural network model, such as a long short-term memory (LSTM) network [45]. As they model time-varying patterns, LSTM networks are commonly used when the temporal changes in a signal are important. In our application, an LSTM network might have avoided the problem of sudden changes in the estimated phase without requiring a voting window. However, in our initial test, the estimation time of LSTM in Raspberry Pi 3B was 0.013 s per sample (estimation rate 73 Hz), lower than the sampling frequency of the inertial sensor (100 Hz), and is therefore unsuitable for estimation in real time. In contrast, the estimation time of the proposed model was 2.45 ms per sample (estimation rate 408 Hz), four times higher than the sampling rate of the sensors. Therefore, phase estimation in our method can be performed in real time.

### 4.2. Cane Orientation Estimation Results

The cane orientation was estimated by the three methods described in Section 2: gravity and magnetic north (Acc + Mag) measurements [15,34], indirect Kalman filter [36], and a Madgwick filter [42]. Table 4 summarizes the root mean squared errors (RMSE) in the 3D cane orientations estimated by the three methods. The Madgwick filter yielded the smallest RMSE (near or below 1°) in all subjects. On average, the RMSE of the Madgwick filter-based method was 19.7 and 1.76 times smaller than those of the Acc + Mag method and indirect Kalman filter, respectively.

The Acc + Mag method obtained the worst results overall. The significantly higher RMSE values of the Acc + Mag method were expected, as the 3D orientation in this method is estimated only from the accelerometer and magnetometer data. Consequently, this method is highly sensitive to the noise in the accelerometer and magnetometer. The results of the indirect Kalman filter were slightly worse than those of the Madgwick filter, which agreed with the results of the original Madgwick filter [42].

Table 5 compares the cane orientations in each activity estimated by the Madgwick filter. The RMSE was lowest in the sit-to-stand and stand-to-sit movements, possibly because these activities were short in duration (only 2.82 s on average) and required minimal movements of the cane. Although the cane was similarly still during the standing activity, the RMSE value of this activity was higher than during the sit-to-stand and stand-to-sit activities. This can be explained by the longer duration (10.5 s on average) of the standing activity, as drift increases the orientation error over time.

The RMSEs of the stair-climbing activities were similar to those of standing, despite the more significant cane movements during the stairs-up and stairs-down activities. The similar RMSE values, despite that the cane moved during the stairs-up and stairs-down activities, were probably attributable to the short durations of these activities (average duration 3.77 s). In contrast, the gait-related activities with both dynamic movements and longer duration (9.12 s on average) yielded the highest RMSE values.

As an example, Figure 8 shows the average pitch angles estimated by the Madgwick filter and their confidence intervals. The absolute error in the pitch angle was compared with the motion capture data for each step of the 2- and 3-point gait activities. The absolute orientation error was larger during the no-contact phase than during the ground-contact phase, because during the no-contact phase, the cane both rotates and translates through the air; accordingly, the acceleration measurements are contributed by both gravitational and movement effects.

Finally, Table 6 compares the average RMSEs in the measurements of each IMU on the cane used by each subject. The estimation method was the Madgwick filter. In all subjects except subject 10, IMU 2 yielded the lowest RMSE. After performing a Kruskal–Wallis test with a significance level of 0.05, it was found that there was significant difference among the results of the inertial sensors.

To determine which IMU’s RMSE significantly differed from those of the other IMUs, a post-hoc analysis based on Dunn’s test [46] was performed. At the 0.05 significance level, the results of IMU 2 significantly differed from those of the other IMUs. However, the sensor position and orientation RMSEs were not obviously related, because IMU 1 and IMU 4 yielded significantly better results than IMU 3 and IMU 5, but significantly worse results than IMU 2.

In summary, although IMU 2 (placed at approximately 20 cm below the handle) achieved the best orientation results, no clear relationship emerged between the sensor position on the cane and the orientation estimation performance. Meanwhile, the Madgwick filter obtained more accurate estimations than the other tested algorithms. As the Madgwick filter requires fewer computations than Kalman filters, it is more suitable for estimating the orientation in real time. As a verification exercise, the Madgwick filter was implemented in Python and tested on a Raspberry Pi 3B single-board computer. The Madgwick filter processed each sample with a duration of 1.26 μs, confirming that the proposed system can estimate cane movements in real time with an overall runtime (phase and orientation runtimes) of 2.45 ms.

## 5. Conclusions

This study proposed a real-time system for estimating the movements of an instrumented cane. In the proposed system, a single inertial sensor is attached to the tip of a normal cane. As the cane is unmodified apart from the sensor attachment, the system is highly usable. The main contributions of this paper are as follows:A deep learning model is proposed for estimating the cane contact-phase estimation from raw inertial sensor data.The temporal phase estimation is improved using a voting window.The 3D orientation of the cane during multiple daily life activities is estimated by orientation estimation algorithms, which are compared and validated.The position effect of the inertial sensor on the cane on both the contact phase and orientation estimates of the cane are analyzed.The proposed system performs the estimation calculations in real time on devices with low computational power.

A main limitation of the present study was the demographic of the participants: all subjects were young with no mobility impairment. However, as the activities of elderly users with mobility impairments are close to stationary activities, the accuracy of the orientation estimates is expected to increase in this group of users. Nevertheless, similar experiments on elderly patients are required to analyze their gait and balance while using the instrumented cane. The experimental data would also reveal how the different gait patterns of elderly participants with different mobility impairments affect the proposed system, and whether the system can detect these differences.

Moreover, the measurements in the second experiment were acquired over short duration because the measurement space of the optical motion capture system was limited. However, a main source of error in inertial sensors is the drift error, which accumulates the orientation estimation error over time. Although the Madgwick filter reduces the drift by fusing the gyroscope data with accelerometer and magnetometer data, longer duration experiments will be conducted in future to analyze the effect of drift on the estimated orientation.

Finally, although the gait can be analyzed using the output parameters of our proposed system, as in previous studies [14], extracting more gait parameters would provide more information on the users’ movements. For example, parameters such as cane stride length, cane clearance, and cane reaction force would help in assessing the gait and balance of users. In future works, parameters such as stride length could be extracted using an indirect Kalman filter, as in [17]. In addition, although the force also provides useful information, installing a force sensor limits the usability of the system. Therefore, in future work, we will estimate the reaction force of the cane by applying a deep learning model to the inertial sensor data.

In summary, the proposed system can estimate both the cane contact-phase and cane orientation in real time. Such a system could provide useful information on the gait and balance of users by extracting the gait parameters, such as cane contact duration, the contact versus noncontact phase duration relationship, or the cane orientations during specific gait events. In future, these parameters extracted from the sensors could be combined with patient record information, such as age, gender, or previous history of fall, in order to generate a more accurate fall risk estimation.

## Figures and Tables

**Figure 1 sensors-20-04675-f001:**
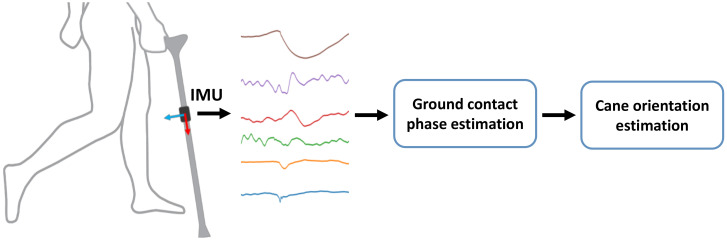
Workflow of the proposed system for estimating the ground-contact phase and 3D cane orientation from inertial sensor data (IMU = inertial measurement unit).

**Figure 2 sensors-20-04675-f002:**
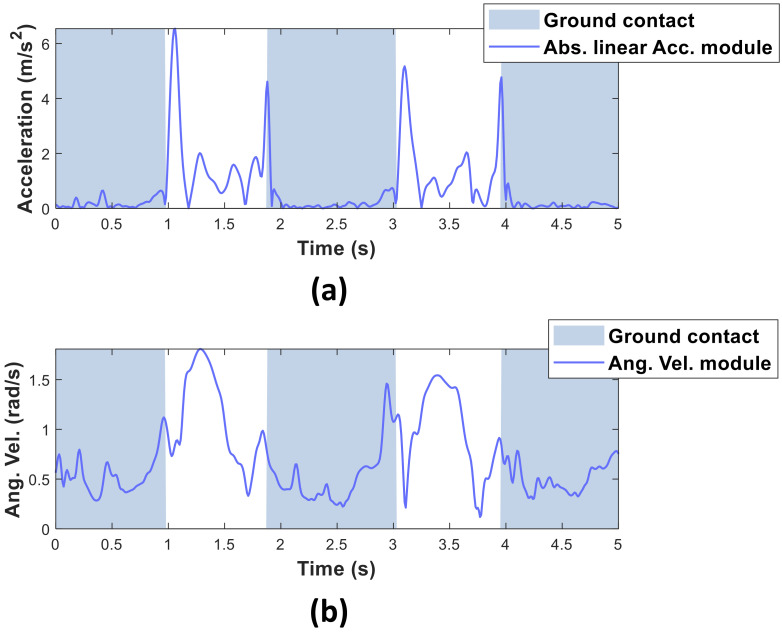
Absolute linear acceleration module (**a**) and angular velocity module (**b**) during ambulation of a single-tip cane user.

**Figure 3 sensors-20-04675-f003:**
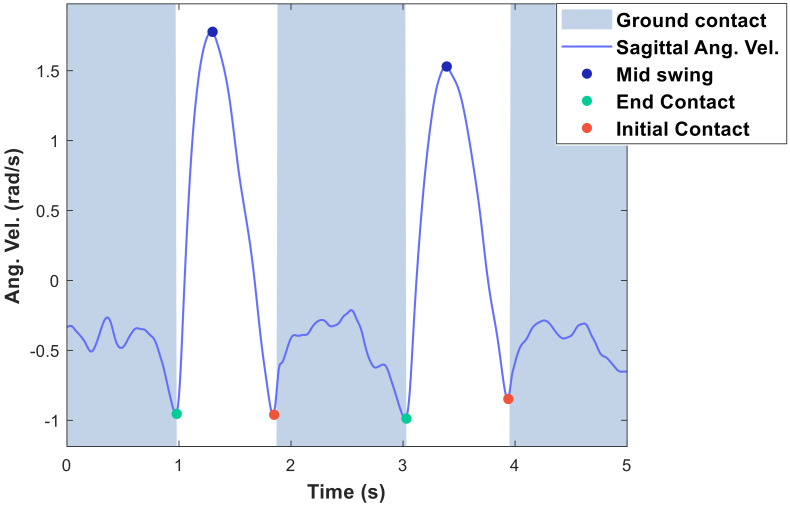
The peaks in the sagittal angular velocity of the cane corresponding with the end contact (green points), mid-swing (blue points), and initial contact (red points) of the cane.

**Figure 4 sensors-20-04675-f004:**
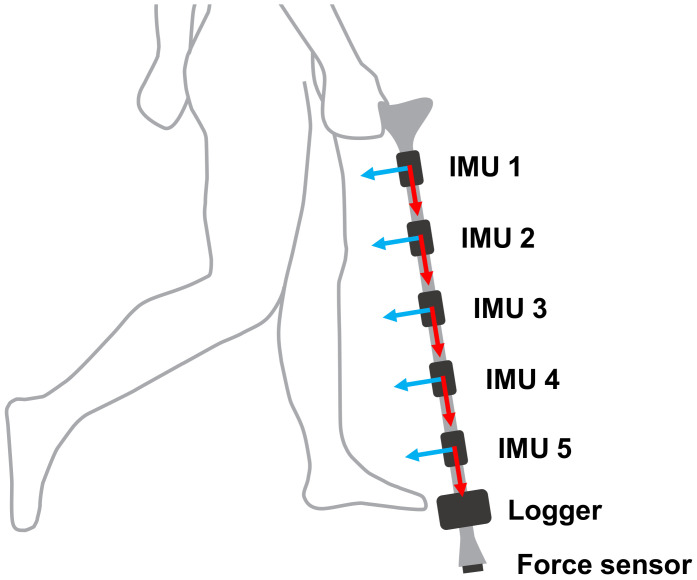
Diagram of the instrumented cane used in this study. Five inertial measurement units (IMU) are placed at equal intervals throughout the cane, and one data logger receives the signal from the force sensor placed on the tip of the cane.

**Figure 5 sensors-20-04675-f005:**
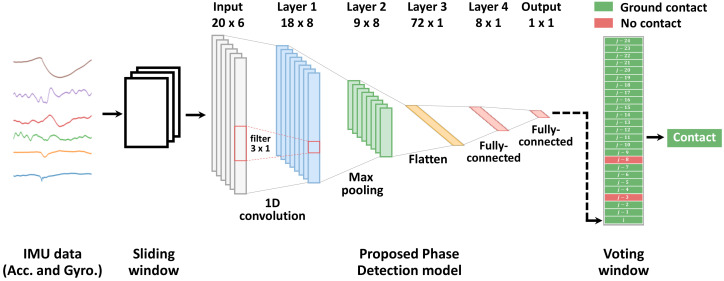
Diagram of the proposed phase estimation. First the sliding window fixes the size of the IMU data to that of the CNN model input. Next, the CNN model estimates the current phase. Finally, the current phase is decided by a voting window enclosing the last 25 estimations.

**Figure 6 sensors-20-04675-f006:**
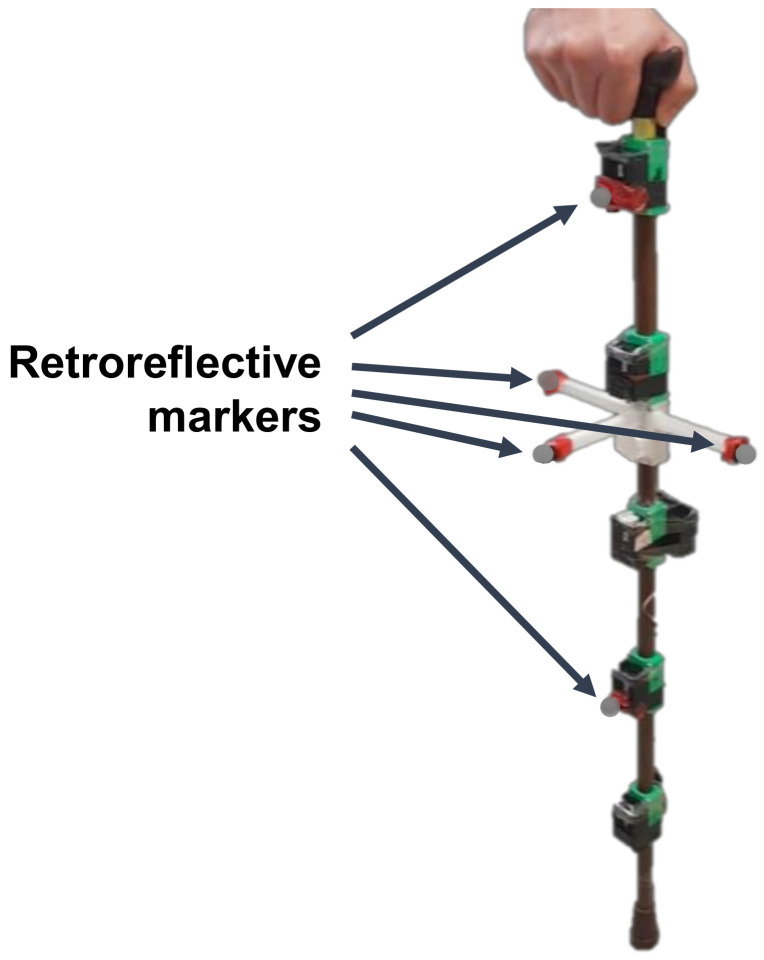
Image of the instrumented cane fitted with retroreflective markers for the optical motion capture system.

**Figure 7 sensors-20-04675-f007:**
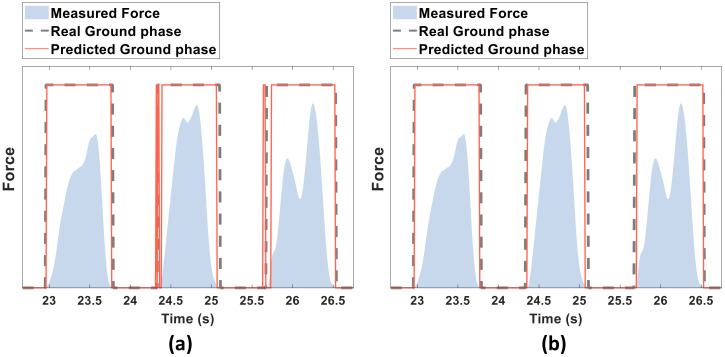
Example of the cane contact phase estimated by the proposed model before (**a**) and after (**b**) applying the voting window.

**Figure 8 sensors-20-04675-f008:**
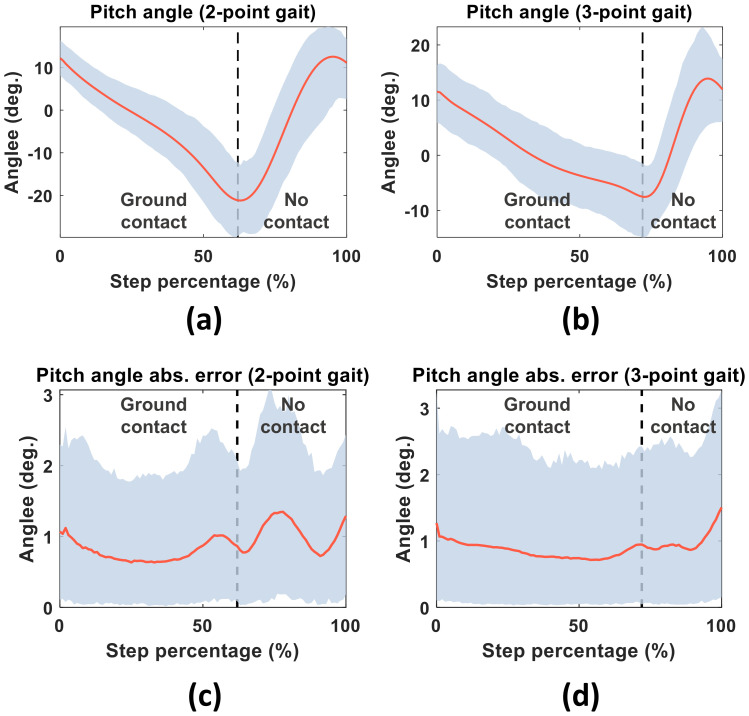
Average pitch values (red curves) and their confidence intervals (blue regions) in the 2-point (**a**) and 3-point (**b**) gaits. Bottom panels show the absolute pitch-angle errors in the 2-point (**c**) and 3-point (**d**) gaits.

**Table 1 sensors-20-04675-t001:** Summary of the activities performed in the experiment.

Activity	Summary
Three-point gait	Walking straight for 3 m following the three-point gait method with the cane. At each step, the participant first moved the cane forward, then moved the left foot forward, and finally moved the right foot forward.
Two-point gait	Walking straight for 3 m following the two-point gait method with the cane. At each step, the participant moved both the cane and the left foot forward, and then moved the right foot forward.
Standing	The participant stood still while bearing weight on the cane for 10 s.
Sit to Stand	The participant moved from a sitting position to a standing position while bearing weight on the cane.
Stand to Sit	The participant moved from a standing position to a sitting position while bearing weight on the cane.
Stairs up	The participant climbed up an 18-cm-high concrete block with a horizontal surface area of (60 × 30) cm^2^. The participant first placed the cane on top of the concrete block, moved the right foot to the top of the block, and finally moved the left foot onto the block.
Stairs down	The participant climbed down from the 18-cm-high concrete block. The participant first brought the cane down to the ground, then brought the left foot down, and finally lowered the right foot to the ground.

**Table 2 sensors-20-04675-t002:** Mean (standard deviation) test accuracies of different cane contact-phase estimation methods.

Method	Subject Number					
	1	2	3	4	5	Avg.
Threshold-based method [17]	83.36	79.17	83.57	82.03	83.21	82.27
	(4.50)	(1.94)	(1.49)	(1.65)	(2.24)	(3.06)
Event-based method [20,21]	79.19	87.79	89.52	83.37	81.80	84.33
	(4.14)	(2.05)	(1.78)	(1.07)	(2.78)	(4.60)
Proposed model	92.77	94.53	96.41	97.32	96.26	95.46
	(3.24)	(1.38)	(1.44)	(0.48)	(1.42)	(2.43)
Proposed model + Voting window	**92.97**	**94.85**	**96.66**	**97.61**	**96.49**	**95.72**
	**(3.22)**	**(1.30)**	**(1.39)**	**(0.42)**	**(1.38)**	**(2.42)**

**Table 3 sensors-20-04675-t003:** Comparison of the mean (standard deviation) test accuracies of the different inertial sensors.

IMU Number	Subject Number					
	1	2	3	4	5	Avg.
IMU 1	92.38	94.04	95.65	96.97	96.09	95.02
	(3.31)	(1.66)	(1.48)	(0.29)	(1.30)	(2.44)
IMU 2	92.83	94.66	96.29	97.45	96.33	95.51
	(3.26)	(1.43)	(1.51)	(0.33)	(1.27)	(2.39)
IMU 3	93.00	94.79	96.60	**97.73**	**96.40**	95.70
	(3.25)	(1.30)	(1.58)	**(0.27)**	**(1.36)**	(2.41)
IMU 4	**93.12**	**95.15**	**96.86**	97.47	96.39	**95.80**
	**(3.19)**	**(1.10)**	**(1.27)**	(0.41)	(1.57)	**(2.30)**
IMU 5	93.08	94.39	96.74	97.22	96.17	95.52
	(3.18)	(1.39)	(1.36)	(0.48)	(1.68)	(2.36)

**Table 4 sensors-20-04675-t004:** Means (standard deviations) of the RMSEs in different methods for estimating the roll, pitch, yaw and total orientation of a cane (unit: degrees).

	Algorithm	Subject Number										
		1	2	3	4	5	6	7	8	9	10	Avg.
Roll	Acc + Mag [15,34]	17.39 (12.93)	17.16 (18.54)	14.68 (15.54)	15.66 (14.87)	17.40 (12.59)	14.44 (12.27)	18.53 (16.71)	26.39 (25.55)	17.25 (14.29)	20.65 (18.83)	17.91 (16.91)
	Indirect Kalman filter [36]	1.55 (1.77)	1.50 (1.28)	0.99 (0.78)	1.03 (0.88)	1.05 (0.64)	1.15 (0.78)	1.29 (1.16)	1.32 (1.15)	1.34 (1.10)	1.19 (0.90)	1.24 (1.10)
	Madgwick filter [42]	**0.67 (0.51)**	**0.68 (0.41)**	**0.66 (0.43)**	**0.75 (0.57)**	**0.87 (0.54)**	**0.70 (0.34)**	**0.67 (0.40)**	**0.67 (0.33)**	**0.88 (0.92)**	**0.77 (0.45)**	**0.73 (0.53)**
Pitch	Acc + Mag [15,34]	16.24 (14.19)	14.81 (15.82)	12.65 (15.62)	11.48 (9.93)	14.78 (11.37)	13.89 (12.29)	16.87 (16.88)	22.65 (25.88)	15.48 (13.30)	18.58 (18.84)	15.71 (16.20)
	Indirect Kalman filter [36]	1.77 (2.10)	1.52 (1.53)	1.46 (1.09)	1.13 (0.66)	1.33 (0.90)	1.25 (1.03)	1.46 (1.64)	1.25 (0.93)	1.49 (1.01)	1.29 (1.00)	1.40 (1.26)
	Madgwick filter [42]	**0.92 (0.59)**	**1.00 (0.71)**	**0.86 (0.45)**	**0.81 (0.35)**	**1.06 (0.54)**	**0.91 (0.53)**	**0.97 (0.62)**	**0.75 (0.38)**	**1.06 (0.59)**	**0.89 (0.49)**	**0.92 (0.54)**
Yaw	Acc + Mag [15,34]	16.91 (12.32)	16.81 (11.02)	13.87 (10.92)	11.96 (8.09)	14.38 (9.81)	14.57 (10.99)	16.04 (10.57)	18.46 (12.18)	14.21 (10.52)	16.96 (10.82)	15.40 (10.92)
	Indirect Kalman filter [36]	1.57 (1.84)	2.12 (3.27)	1.25 (1.31)	1.01 (0.89)	1.25 (1.33)	1.26 (1.60)	2.09 (2.92)	1.20 (1.11)	1.49 (1.71)	1.60 (2.37)	1.48 (2.01)
	Madgwick filter [42]	**0.71 (0.53)**	**0.68 (0.44)**	**0.70 (0.41)**	**0.71 (0.48)**	**0.84 (0.49)**	**0.71 (0.43)**	**1.15 (0.57)**	**0.78 (0.46)**	**0.91 (0.93)**	**0.67 (0.40)**	**0.78 (0.55)**
Total	Acc + Mag [15,34]	17.45 (12.36)	17.25 (14.36)	14.35 (13.59)	13.98 (10.30)	16.15 (10.48)	14.92 (11.08)	17.93 (14.11)	23.74 (21.06)	16.31 (12.02)	19.61 (15.62)	17.14 (14.04)
	Indirect Kalman filter [36]	1.84 (1.71)	1.97 (2.01)	1.37 (0.92)	1.17 (0.65)	1.32 (0.86)	1.34 (1.05)	1.83 (1.89)	1.37 (0.92)	1.60 (1.11)	1.53 (1.42)	1.53 (1.35)
	Madgwick filter [42]	**0.82 (0.47)**	**0.85 (0.46)**	**0.78 (0.36)**	**0.81 (0.38)**	**0.96 (0.45)**	**0.81 (0.38)**	**1.00 (0.43)**	**0.78 (0.29)**	**1.03 (0.74)**	**0.83 (0.36)**	**0.87 (0.46)**

**Table 5 sensors-20-04675-t005:** Means (standard deviations) of the RMSEs in the roll, pitch, yaw and total cane orientations estimated by the Madgwick filter (unit: degrees).

	Activity Type						
	3-Point Gait	2-Point Gait	Standing	Sit to Stand	Stand to Sit	Stairs Up	Stairs Down
Roll	1.13 (0.89)	1.03 (0.44)	0.80 (0.54)	0.45 (0.21)	0.44 (0.18)	0.53 (0.23)	0.51 (0.22)
Pitch	1.08 (0.39)	1.00 (0.26)	0.74 (0.46)	0.52 (0.28)	0.65 (0.36)	0.92 (0.46)	0.49 (0.20)
Yaw	1.14 (0.86)	0.93 (0.42)	0.98 (0.56)	0.64 (0.34)	0.68 (0.38)	0.67 (0.52)	0.94 (0.47)
Total	1.24 (0.60)	1.09 (0.26)	0.86 (0.40)	0.57 (0.19)	0.56 (0.23)	0.81 (0.34)	0.73 (0.21)

**Table 6 sensors-20-04675-t006:** Comparison of mean (standard deviation) RMSEs in the cane orientation of the different IMU sensors on the cane (unit: degrees; estimation method: Madgwick filter).

	Subject Number										
	1	2	3	4	5	6	7	8	9	10	Avg.
IMU 1	0.77 (0.43)	0.81 (0.46)	0.73 (0.31)	0.85 (0.33)	1.02 (0.51)	0.79 (0.36)	1.14 (0.56)	0.78 (0.27)	0.97 (0.73)	**0.68 (0.33)**	0.85 (0.47)
IMU 2	**0.63 (0.36)**	**0.57 (0.29)**	**0.53 (0.23)**	**0.66 (0.35)**	**0.82 (0.33)**	**0.73 (0.31)**	**0.81 (0.29)**	**0.72 (0.28)**	**0.92 (0.69)**	0.71 (0.27)	**0.71 (0.38)**
IMU 3	1.03 (0.56)	0.99 (0.50)	0.84 (0.33)	0.84 (0.36)	1.08 (0.52)	0.91 (0.46)	1.01 (0.41)	0.78 (0.30)	1.11 (0.75)	1.03 (0.39)	0.96 (0.49)
IMU 4	0.77 (0.41)	0.82 (0.37)	0.76 (0.32)	0.76 (0.39)	0.90 (0.36)	0.76 (0.31)	0.94 (0.36)	0.77 (0.30)	1.05 (0.74)	0.85 (0.33)	0.79 (0.54)
IMU 5	0.90 (0.44)	1.05 (0.47)	1.03 (0.39)	0.96 (0.39)	0.99 (0.46)	0.87 (0.43)	1.11 (0.43)	0.86 (0.30)	1.09 (0.78)	0.88 (0.35)	0.97 (0.47)

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
