# Peer review of "Inertial Sensor-Based Instrumented Cane for Real-Time Walking Cane Kinematics Estimation"

_sensors, 2020, doi:10.3390/s20174675_

Round 1
Reviewer 1 Report
The authors presented the idea of cane equipped with inertial sensors to predict fall. The idea is interesting and might be useful to the aging society. The methods are introduced with details and the results look promising.
A few suggestions which may further improve the manuscript:
1. Page 1, line 31, "ageing" should be "aging".
2. The authors used "falling" for sometime and "fall" for sometime when they referred to the same thing. Preferably use "fall" and be consistent.
3. No unit for x-axis in Figure. 2 and 3.
4. What is the reason to use 5 IMUs? Since the cane is a rigid body, is it redundant? Will it affect the user by adding extra weights?
5. Extensive edit is needed to polish the language.
6. No unit for y-axis in Figure. 7.
Author Response
Response to Reviewer 1 Comments
Point 1: Page 1, line 31, "ageing" should be "aging".
Response 1: The word “ageing” has been replaced with the word “aging” as suggested.
Point 2: The authors used "falling" for sometime and "fall" for sometime when they referred to the same thing. Preferably use "fall" and be consistent.
Response 2: All the instances of the word “falling” have been replaced with the word “fall” as suggested.
Point 3: No unit for x-axis in Figure. 2 and 3.
Response 3: The x-axis of both figures has been modified to display the time axis.
Point 4: What is the reason to use 5 IMUs? Since the cane is a rigid body, is it redundant? Will it affect the user by adding extra weights?
Response 4: Even though the cane is a rigid body and that all the IMU measure a similar angular velocity, since the cane moves like a pendulum, there is a difference in the accelerations due to the rotation of the cane. For example, the IMU closest to the contact point of the cane with the ground has a lower velocity than the IMU on the opposite side with a higher rotation radius.
Regarding the weight of the IMU, each sensor weights around 30 g, so even using 5 IMU the weight of the cane only increases to 150 g, which should not affect to the user. Moreover, in the final system that will be used by the elderly, only one IMU will be placed in the cane.
Point 5: Extensive edit is needed to polish the language.
Response 5: We have corrected the language suggested in the previous points. Also, the original manuscript was edited for English language, grammar, punctuation, and spelling by an English Editing Company. Please see the attachment with the certificate of editing. If there is any specific language mistake that needs to be fixed, we would appreciate to hear about it.
Point 6: No unit for y-axis in Figure. 7.
Response 6: The label for the y-axis in Figure 7 has been added.

Reviewer 2 Report
Your reference of WHO is too old. The WHO has published a new report. Currently the draft is accessible on the web and should be released. It holds some new implications.
Your references on falls incidence Japan and Malaysia are taken. Please explain why the falls incidence is lower in Japan than US/Europe. I.E. body height, ADL differences, etc
The use of walking aids differs world wide (e.g. cane, rollators). Please add some input on cross cultural data. The design and ergonomics vary widely e.g in Scandinavia and US.
A missing bit from my side is the option of hybrid or fusion using data from the person and the cane.
In summary, from a clinical perspective a novel and interesting paper. Of course we need real world data from older subjects.
Thank you for this interesting work
Author Response
Response to Reviewer 2 Comments
Point 1: Your reference of WHO is too old. The WHO has published a new report. Currently the draft is accessible on the web and should be released. It holds some new implications.
Response 1: We could not find the mentioned draft; we would appreciate to know where the mentioned draft is available. However, we have updated the reference to the “World report on ageing and health” published in 2015.
Point 2: Your references on falls incidence Japan and Malaysia are taken. Please explain why the falls incidence is lower in Japan than US/Europe. I.E. body height, ADL differences, etc
Response 2: We have added information on this regard in the first paragraph on page 1 based on the data in “Incidence of Falls among the Elderly and Preventive Efforts in Japan” (Yasumura S. et al. 2009), as follows:
“Although this trend might be caused by a difference in the methods for surveying falls, it is thought that the difference in the fall incidence might be caused by dietary and lifestyle habits, which are also related with a higher life expectancy [5].
Point 3: The use of walking aids differs world wide (e.g. cane, rollators). Please add some input on cross cultural data. The design and ergonomics vary widely e.g in Scandinavia and US.
Response 3: We have not been able to find any study comparing the design and ergonomics of walking aids among different countries. However, we have modified the second paragraph in the second page (line 56) to include information about the use of walking aids worldwide, as follows:
“For this reason, many elderly individuals are provided with assistive devices such as walkers and walking canes, being the later the most common prescribed mobility aid worldwide [9,10]”
Point 4: A missing bit from my side is the option of hybrid or fusion using data from the person and the cane.
Response 4: We have modified the last paragraph of the Conclusions section with the idea that the fall risk estimation could be further improved if the sensor data is combined with the patient information (age, gender or previous fall history), as follows:
“In future, these parameters extracted from the sensors could be combined with patient record information, such as age, gender or previous history of fall, in order to generate a more accurate fall risk estimation.”